# In Vitro and In Vivo Characterization of ^89^Zirconium-Labeled Lintuzumab Molecule

**DOI:** 10.3390/molecules27196589

**Published:** 2022-10-05

**Authors:** Kevin J. H. Allen, Rubin Jiao, Jason Li, Denis R. Beckford-Vera, Ekaterina Dadachova

**Affiliations:** 1College of Pharmacy and Nutrition, University of Saskatchewan, Saskatoon, SK S7N 5E5, Canada; 2Actinium Pharmaceuticals, New York, NY 10016, USA

**Keywords:** lintuzumab, 89Zirconium, positron emission tomography, acute myeloid leukemia

## Abstract

**Objective:** Positron emission tomography (PET) imaging is a powerful non-invasive method to determine the in vivo behavior of biomolecules. Determining biodistribution and pharmacokinetic (PK) properties of targeted therapeutics can enable a better understanding of in vivo drug mechanisms such as tumor uptake, off target accumulation and clearance. Zirconium-89 (^89^Zr) is a readily available tetravalent PET-enabling radiometal that has been used to evaluate the biodistribution and PK of monoclonal antibodies. In the current study, we performed in vitro and in vivo characterization of ^89^Zr-lintuzumab, a radiolabeled anti-CD33 antibody, as a model to evaluate the in vivo binding properties in preclinical models of AML. **Methods:** Lintuzumab was conjugated to p-SCN-Bn-deferoxamine (DFO) and labeled with ^89^Zr using a 5:1 µCi:µg specific activity at 37 °C for 1h. The biological activity of ^89^Zr-lintuzumab was evaluated in a panel of CD33 positive cells using flow cytometry. Fox Chase SCID mice were injected with 2 × 10^6^ OCI-AML3 cells into the right flank. After 12 days, a cohort of mice (*n* = 4) were injected with ^89^Zr-lintuzumab via tail vein. PET/CT scans of mice were acquired on days 1, 2, 3 and 7 post ^89^Zr-lintuzumab injection. To demonstrate ^89^Zr-lintuzumab specific binding to CD33 expressing tumors in vivo, a blocking study was performed. This cohort of mice (*n* = 4) was injected with native lintuzumab and 24 h later ^89^Zr-lintuzumab was administered. This group was imaged 3 and 7 days after injection of ^89^Zr-lintuzumab. A full ex vivo biodistribution study on both cohorts was performed on day 7. The results from the PET image and ex vivo biodistribution studies were compared. **Results:** Lintuzumab was successfully radiolabeled with ^89^Zr resulting in a 99% radiochemical yield. The ^89^Zr-lintuzumab radioconjugate specifically binds CD33 positive cells in a similar manner to native lintuzumab as observed by flow cytometry. PET imaging revealed high accumulation of ^89^Zr-lintuzumab in OCI-AML3 tumors within 24h post-injection of the radioconjugate. The ^89^Zr-lintuzumab high tumor uptake remains for up to 7 days. Tumor analysis of the PET data using volume of interest (VOI) showed significant blocking of ^89^Zr-lintuzumab in the group pre-treated with native lintuzumab (pre-blocked group), thus indicating specific targeting of CD33 on OCI-AML3 cells in vivo. The tumor uptake findings from the PET imaging study are in agreement with those from the ex vivo biodistribution results. **Conclusions:** PET imaging of ^89^Zr-lintuzumab shows high specific uptake in CD33 positive human OCI-AML3 tumors. The results from the image study agree with the observations from the ex vivo biodistribution study. Our findings collectively suggest that PET imaging using ^89^Zr-lintuzumab could be a powerful drug development tool to evaluate binding properties of anti-CD33 monoclonal antibodies in preclinical cancer models.

## 1. Introduction

Positron emission tomography (PET) imaging is a powerful non-invasive tool to determine the in vivo behavior of biomolecules in oncology [1,2]. Determining biodistribution and pharmacokinetic (PK) properties of targeted therapeutics can enable a better understanding of in vivo drug mechanisms such as tumor uptake, off target accumulation, and clearance [3]. To obtain a clear understanding of the biodistribution and pharmacokinetics of biomolecules using radiotracers, the physical half-life of the radionuclide should match the biological half-life of the biomolecule. If a radionuclide half-life is significantly shorter than the biological half-life of a biomolecule—the radiolabel will have already decayed away while the biomolecule is still distributed in different body compartments, leading to the loss of valuable pharmacokinetics data. Monoclonal antibodies have a relatively long biological half-life and therefore an isotope with a similar physical half-life is necessary. Currently, two PET enabling radionuclides are routinely available to the nuclear medicine community: Gallium-68 (^68^Ga), a trivalent positron radiometal with a short physical half-life of 68 min and Zirconium-89 (^89^Zr), tetravalent radiometal with a half-life of 78.4 h. ^68^Ga half-life does not match the several days long biological half-life of full-size antibodies; ^89^Zr half-life provides a better match. Recently pre-clinical imaging of acute myeloid leukemia (AML) models with Copper-64 (^64^Cu) has been reported [4], however, both short-physical half-life of ^64^Cu (12.7 h) and its not being readily available limit its use as a research tool for biodistribution and pharmacokinetics of antibodies. It has been demonstrated that targeting CD33 with lintuzumab antibody radiolabeled with ^225^Ac has promising activity in patients with relapsed/refractory AML in phase 1/2 clinical trials [5,6,7]. In the current study, we labeled lintuzumab with ^89^Zr and performed in vitro and in vivo characterization of this molecule with the ultimate goal to develop a preclinical tool to study CD33 tumor targeting in AML models using PET. 

## 2. Results

### 2.1. Conjugation and Radiolabeling of Lintuzumab with ^89^Zr Resulted in Preservation of Lintuzumab Immunoreactivity and Quantitative Yields

The immunoreactivity of lintuzumab conjugated to p-isothiocyanato-benzyl-desferrioxamine (p-SCN-Bz-DFO) was evaluated by ELISA and flow cytometry. The binding of the conjugate to recombinant human CD33 protein was identical to that of native lintuzumab as was demonstrated via ELISA (Figure 1A). There was no binding of control human IgG to CD33 antigen. Flow cytometry of lintuzumab-DFO binding to CD33 expressing human cancer cell lines MV411, HL60, and U937 revealed that the conjugate preserved at least 85% of cold lintuzumab immunoreactivity (Figure 1B). These results provided the impetus for the in vivo characterization of ^89^Zr-lintuzumab.

Radiolabeling of lintuzumab with ^89^Zr resulted in quantitative radionuclide incorporation for both the 0.185 MBq/1 µg and 0.37 MBq/1 µg specific activities. High performance liquid chromatography (HPLC) analysis of the ^89^Zr-lintuzumab radiolabeled at 0.185 MBq/1 µg specific activity showed a single peak in the radiochromatogram (Figure 1C) which was associated with the antibody, and that there were no signs of protein degradation in the UV trace of the antibody. Radio instant thin layer chromatography (iTLC) confirmed that a radiolabeling yield of greater than 99% was achieved after 1 h incubation at 37^o^C, and no post-labeling purification was required. The 0.185 MBq/1 µg was chosen for all follow-up experiments.

### 2.2. In Vivo Imaging and Ex Vivo Biodistribution Demonstrated Specific Uptake of ^89^Zr-Lintuzumab in the OCI-AML3 Tumors and Fast Clearance from Circulation

Four OCI-AML-3 tumor bearing mice were administered 5.55 MBq ^89^Zr-DFO-Lintuzumab and microPET/CT mages were taken on Days 1, 2, 3, and 7 post administration. The radioconjugate cleared from the blood and other organs except for the tumor after Day 1 and accumulated in CD33-positive tumors (Figure 2A). There was some residual activity in the heart on Days 2 and 3, which disappeared by Day 7. Volume of interest (VOI) analysis (decay corrected) of microPET images showed that the amount of antibody remained relatively constant in the tumors for at least 72 h after injection.

To demonstrate ^89^Zr-lintuzumab specific binding to CD33 expressing tumors in vivo, a blocking study was performed. A cohort of four OCI-AML-3 tumor bearing mice pre-blocked with “cold” (unlabeled) lintuzumab 24 h prior to administration of radioconjugate was imaged with microPET/CT on Days 3 and 7 after radioconjugate administration. The clearance of radioconjugate from blood and other organs in mice pre-blocked with lintuzumab was not as effective as in non-blocked mice: radioactivity was detected in circulation and various organs even on Day 7 after administration (Figure 2B). SUV analysis of tumor images taken on Days 3 and 7 showed a significantly (*p* = 0.001 and *p* < 0.0001, respectively) higher uptake of ^89^Zr-DFO-Lintuzumab in the tumors of non-blocked mice than in the pre-blocked tumors (Figure 2C). In addition, time–activity curves (TAC) show that the radiolabeled antibody remains mostly constant in the tumor over the 7-day period after initial uptake for both non-blocked and pre-blocked tumors, with the pre-blocked tumors having significantly less uptake (Figure 2D).

After imaging was complete both cohorts were sacrificed at 168 h post ^89^Zr-lintuzumab and an ex vivo biodistribution was performed to compare organ uptake with the results obtained using VOI analysis. There were significant differences between the blood and tumor uptakes in pre-blocked and non-blocked cohorts (*p* = 0.0017 and *p* = 0.0014, respectively) with higher uptake being observed in the blood of pre-blocked mice (Figure 3A), and conversely, lower uptake in their tumors (Figure 3B). Overall, imaging and biodistribution results on day 7 showed good agreement with each other with tumor uptake being 1.71 (microPET/CT) and 1.99 (biodistribution) times higher in non-blocked cohort than in pre-blocked mice. 

## 3. Discussion

One important aspect of preclinical drug development is the ability to evaluate tumor targeting in vivo. PET is a very powerful non-invasive technique that can be used to assess real time tumor targeting in vivo in preclinical models of different diseases. Developing methods to utilize this technique can be very beneficial in generating data such as in vivo target expression and distribution that is not possible with other approaches. While AML is being successfully diagnosed with blood tests followed by bone marrow biopsies [8], an in vivo research tool to study the interaction of the anti-CD33 antibodies with the tumors would be beneficial. For example, the timing of an anti-CD33 antibody uptake, retention and washout from the tumor will determine the timing of administration of an additional anti-cancer agent(s) for combination therapy to achieve a synergistic effect. Here, we report the results of in vitro and in vivo characterization of ^89^Zr-lintuzumab as a model to evaluate the in vivo binding properties of anti-CD33 antibodies in preclinical models of CD33 expressing AML using PET.

Lintuzumab was successfully radiolabeled with ^89^Zr resulting in a 99% radiochemical yield. This is in contrast with the values reported by Buckway et al. who observed approximately 58% radiolabeling yields of DFO-conjugated lintuzumab with ^89^Zr which was subsequently used for binding to CD33 positive HL-60 tumors in mice [9]. This difference might be due to different buffers used for the radiolabeling procedure—HEPES was utilized in this work, while PBS was used in [9]. We did not determine the serum stability of ^89^Zr-DFO-lintuzumab, as such stability is determined by the stability of ^89^Zr-DFO complex, and its stability has been previously demonstrated in several studies [10,11,12]. Some uptake in the bone marrow observed in mice is due not to the instability of the radioconjugate but to the binding of humanized Fc part of lintuzumab to the murine FcRn receptors which is known to be much stronger for humanized than for murine antibodies [13]. The ^89^Zr-lintuzumab radioconjugate specifically bound CD33 positive cells in a similar manner to native lintuzumab as observed by flow cytometry. microPET imaging revealed high accumulation of ^89^Zr-lintuzumab in OCI-AML3 tumors within 24 h post-injection of the radioconjugate. The ^89^Zr-lintuzumab high tumor uptake remained for up to 7 days. Tumor analysis of the microPET data using VOIs showed significant blocking of ^89^Zr-lintuzumab in the group pre-treated with naive lintuzumab (pre-blocked group), thus indicating specific targeting of CD33 on OCI-AML3 cells in vivo. Pre-blocking of uptake with the unlabeled specific molecule administration is a well-tested way to demonstrate the specificity of a molecule for its receptor or an antigen [14]. Analysis of the time activity curve (TAC) shows that there is only a slight downwards slope, indicating that the majority of the ^89^Zr that accumulates in the tumor remains present there until decay. The slower clearance of the radiolabeled antibody from the circulation in pre-blocked animals can be explained by its inability to bind in high amount to the pre-blocked tumor (which usually serves as an antibody “sink”), thus leaving the majority of the radiolabeled antibody molecules in the circulation for a long time. The tumor uptake findings from the microPET imaging study were confirmed by the ex vivo biodistribution results. The tumor to blood and tumor to muscle ratios were in agreement between PET and biodistribution on day 7—for example, in pre-blocked mice tumor to blood ratio by PET was 0.72 and by biodistribution – 0.65, tumor to muscle ratio by PET was 4.7 and by biodistribution – 4.0. 

## 4. Materials and Methods

Cell lines: Human AML cell line OCI-AML3 (DSMZ No.: ACC 582) was purchased from Deutsche Sammlung von Mikroorganismen und Zellkulturen (DSMZ) and cultured in RPMI-1640 (Cat no: SH30255.01, Thermo Fisher Scientific, Waltham, MA, USA) supplemented with 10% fetal bovine serum (Sigma-Aldrich, St. Louis, MO, USA) and 1% antibiotic/antimycotic (Cat. No.: 15–240–062, Thermo Fisher Scientific). Cells were kept at 37 °C in a 5% CO_2_ incubator. CD33 positive human cancer cell lines MV411 (CRL-9591), HL60 (CCL-240) and U937 (CRL-3253) were purchased from ATCC and cultured following ATCC guidelines.

Lintuzumab antibody conjugation: Lintuzumab was conjugated to the bifunctional chelator *p*-SCN-Bn-DFO (Macrocyclics, Plano, TX, USA) via modified literature methods [3,15,16,17]. In brief, 500 ug of Lintuzumab was exchanged into carbonate conjugation buffer, pH = 8.5, and conjugated to the chelator using a 3-fold molar excess of *p*-SCN-Bn-DFO over antibody and incubated at 37 °C for 1.5 h. Upon completion of the reaction the lintuzumab-DFO conjugate was washed 10 times with 0.5 M HEPES buffer at 4 °C to remove excess *p*-SCN-Bn-DFO giving a lintuzumab-DFO conjugate. The chelating agent-antibody ratio (CAR) for DFO molecules per antibody was determined via MALDI-TOF (University of Alberta) to be 1.2.

ELISA: Nunc MaxiSorp flat-bottomed 96-well plates were coated with 100 ng/well of human recombinant CD33 His-Tag protein (R&D Systems) in PBS and incubated overnight at 4 °C. The ELISA was performed according to standard protocol. Primary antibody stock solutions of lintuzumb, DFO-lintuzumab and IgG as a control were prepared at a concentration of 100 μg/mL and serially diluted tenfold for a concentration range of (10 ng/mL–100 µg/mL). The dilutions were added to the previously CD33 coated wells, afterwards the signal was developed by adding a secondary antibody (Goat anti-human IgG F(ab’)_2_-HRP) and 1.0 mol/L HCl. Absorbance was measured at 450 nm using a Tecan Infinite M-Plex reader and analyzed using GraphPad Prism 9.4 software (San Diego, CA, USA). Each experiment was done in duplicates and at least two independent experiments were performed.

Flow Cytometry: Solutions of lintuzumb, DFO-lintuzumab and IgG as a control, prepared at a concentration of 100 μg/mL, were incubated with 300,000 human CD33 expressing cells (MV-4-11, U937 or HL-60) cells for 1 h on ice. Cells were stained using a PE mouse anti-human IgG diluted in Stain Buffer FBS (BD Pharmingen) for 1 h on ice. The signal was acquired on a BD Accuri C6 flow cytometer and data analyzed using GraphPad Prism 9.4 software. Each experiment was done in duplicates and at least two independent experiments were performed.

Radiolabeling with ^89^Zr: The desired amount of ^89^Zr(Ox)_2_ in 1M oxalic acid was dissolved in 0.5M HEPES buffer that had been run through a chelex cation exchange resin to remove any advantageous metals and neutralized using 1M Na_2_CO_3_. Lint-DFO was then added to achieve a 0.185 MBq/1 µg and 0.37 MBq/1 µg specific activity. The reaction was quenched using 3 µL of 0.05 M DTPA solution to bind any free ^89^Zr and the percentage of radiolabeling yield measured by instant thin layer chromatography (iTLC) (Agilent Technologies, Santa Clara, CA, USA) using a 2470 Wizard2 Gamma counter (Perkin Elmer, Waltham, MA, USA) calibrated for ^89^Zr emission spectra and a radioHPLC trace (Agilent Technologies, Santa Clara, CA, USA). iTLC were eluted using a 50mM EDTA mobile phase with the radiolabeled antibody remaining at the point of application (Rf = 0) and the DTPA bound ^89^Zr moves with the solvent front (Rf = 1), the strip was then cut in half and the measured individually, allowing for calculation of radiolabeling yields. SEC-HPLC traces were performed using a Tosoh BioScience (Tokyo, Japan) TSKgel UP-SW-3000 SEC column with an isocratic method using phosphate-buffered saline (PBS) as a mobile phase. UV and Radio traces were collected, and the peak area compared to iTLC results, only a single radio peak that corresponded with the antibody UV peak was observed. Radiolabeling yields were greater than 99% for both SA and no further purification was required. 0.185 MBq/1 µg SA was used in the subsequent experiments.

Tumor initiation and ^89^Zr-lintuzumab administration: Fox Chase SCID mice were injected with 2 × 10^6^ OCI-AML3 cells into the right flank as previously described [18]. After 11 days, 8 mice were randomized into 2 cohorts (*n* = 4) with equal tumor distribution. Pre-blocked cohort was injected IV with 0.5 mg of lintuzumab. On Day 12 both cohorts were administered ^89^Zr-lintuzumab via tail vein. 

Small animal microPET/CT imaging and biodistribution of tumor-bearing mice: Scans of non-blocked mice (*n* = 4) were acquired 24, 48, 72 and 168 hrs post ^89^Zr-lintuzumab administration. PET/CT scans of mice pre-blocked with “cold” lintuzumab were collected 72 and 168 h post injection of ^89^Zr-lintuzumab. microPET/CT were collected on a Sofie GNEXT microPet/CT scanner (Sofie, Dulles, VA, USA). Four mice were scanned for 10 min (static) simultaneously using a multi-mouse bed. PET/CT images were registered and reconstructed automatically by the GNEXT system. PET images were reconstructed using a 3D-ordered Subset Expectation Maximization algorithm with 24 subsets and 3 iterations, CT images were reconstructed using a Modified Feldkamp algorithm. Reconstructed PET images were filtered using gaussian smooth 3D FWHM 2 × 2 × 2 mm and analyzed using p-MOD v3.903 (Zurich, Switzerland). Volume of interest (VOI) analysis was performed using p-MOD. MIP images were generated to help visualize tumor uptake. Tumor VOI were drawn manually, activity was decay corrected. Standardized uptake values (SUV) were calculated using the equation SUV = C/(dose/weight) where C is the tissue radioactivity concentration, weight is weight of mouse and dose is injected dose of radioactive antibody. The images were reprocessed and normalized to SUV using the same scale. An ex vivo biodistribution study on both cohorts was performed on day 7. Organs were collected and measured using a 2470 Wizard2 Gamma counter. The results from the microPET images and ex vivo biodistribution studies were then compared. 

Statistical analyses. Power analysis for the PET/CT and biodistribution studies was estimated using PASS version 11 (NCSS, Inc. Kaysville, UT, USA) using simulations of different tumor uptakes based on pilot data and conservative assumptions regarding the groups pre-blocked with unlabeled lintuzumab. All simulations showed power of at least 80% with only four animals per group because of the large differences between uptake in pre-blocked and non-blocked mice. Thus, 4 mice per group were utilized in the in vivo studies. Statistical data was generated using GraphPad Prism 9.4 (San Diego, CA, USA). Student t test was used to analyze if there was significant difference between the groups. 

## 5. Conclusions

PET imaging of ^89^Zr-lintuzumab shows high specific uptake in CD33 positive human OCI-AML3 tumors. The results from the image study agree with the observations from the ex vivo biodistribution study. Our findings collectively suggest that PET imaging using ^89^Zr-lintuzumab could be a powerful pre-clinical drug development tool to evaluate binding properties of anti-CD33 monoclonal antibodies in preclinical cancer models.

## Figures and Tables

**Figure 1 molecules-27-06589-f001:**
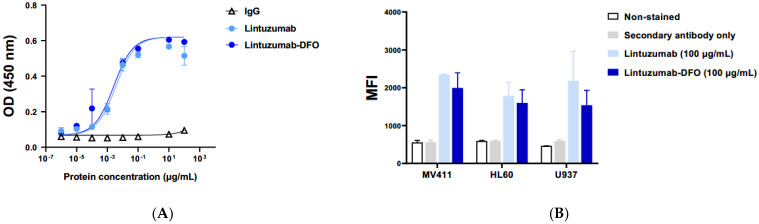
Radiolabeling of lintuzumab with ^89^Zr and its in vitro binding to CD33. (**A**) Binding of Lintuzumab−DFO conjugate to recombinant human CD33 protein is demonstrated via ELISA. (**B**) Binding of Lintuzumab−DFO conjugate to human cancer cell lines that express CD33 is demonstrated via immunofluorescence staining using a flow cytometer. (**C**) HPLC trace chromatograms ran on purified antibody conjugate (upper panel) and radiolabeled conjugate at a wavelength of 280 nm (middle panel UV trace, lower panel radioactivity trace).

**Figure 2 molecules-27-06589-f002:**
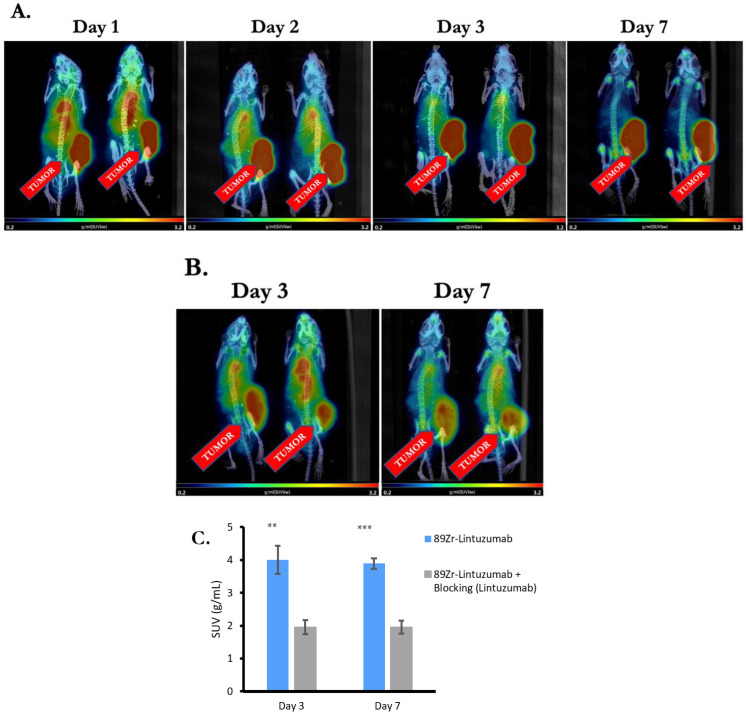
^89^Zr-lintuzumab Selectively Accumulates in CD33 Expressing Tumors. (**A**) PET/CT imaging of mice administered with 5.55 MBq ^89^Zr-DFO-Lintuzumab only (*n* = 4). Images were taken on Days 1, 2, 3, and 7 post administration. The radioconjugate was cleared from blood and other organs except for the tumor after Day 1 and accumulated in CD33-positive OCI-AML-3 tumors. (**B**) Cohort of mice (*n* = 4) pre-blocked with cold Lintuzumab 24hr prior to administration of radioconjugate. PET/CT images were taken on Days 3 and 7 post radioconjugate administration. The clearance of radioconjugate from blood and other organs in mice pre-blocked with Lintuzumab was not as effective as in non-blocked mice: radioactivity was detected in various organs by PET/CT imaging even on Day 7 post administration. (**C**) Standardized uptake values (SUV) analysis of PET/CT images taken on Days 3 and 7. Tumor volumes of interest (VOI) were drawn, and SUV were calculated. ^89^Zr-DFO-Lintuzumab showed a significantly higher uptake in the tumors of non-blocked mice than in the tumors of pre-blocked mice. (**D**) Time–activity curves (TAC) show that the radiolabeled antibody remains mostly constant over the 7-day period after initial uptake for both non-blocked and pre-blocked tumors, with the pre-blocked has significantly less uptake. All images are displayed as maximum intensity projections (MIP). ** and *** mean *p* = 0.001 and *p* < 0.0001, respectively.

**Figure 3 molecules-27-06589-f003:**
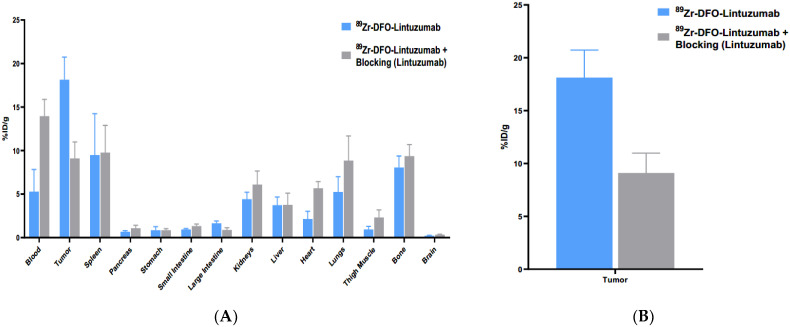
^89^Zr-lintuzumab biodistribution in mice bearing AML-OCI3 tumors. (**A**) Ex vivo biodistribution study of ^89^Zr-DFO-Lintuzumab in mice bearing OCI-AML-3. Mice were inoculated with 2 × 10^6^ OCI-AML3 cells in the right flank. Tumors were allowed to grow for 12 days prior to lintuzumab injection. To demonstrate in vivo target specificity, a cohort of mice (*n* = 4) was blocked with 0.5 mg cold Lintuzumab on Day 0. 5.55 MBq ^89^Zr-DFO-Lintuzumab was administered to both cohorts of mice (blocked and non-blocked; *n* = 4 for each) on Day 1. All mice were subsequently sacrificed on Day 8 post-administration; tumors and several tissues were harvested, weighted, and measured for radioactivity. (**B**) High specific uptake of ^89^Zr-DFO-Lintuzumab in OCI-AML-3 tumors is observed in the cohort of mice without Lintuzumab blocking.

## Data Availability

All data can be found in the manuscript.

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
