# Peer review of "In Vitro and In Vivo Characterization of 89Zirconium-Labeled Lintuzumab Molecule"

_molecules, 2022, doi:10.3390/molecules27196589_

Round 1

Reviewer 1 Report

The submitted article entitled "In vitro and in vivo characterization of 89Zirconium-labeled lintuzumab molecule” evaluates the use of Zr-89 radiometal for PET imaging of lintuzumab. The use of Zr-89 provides a longer half-life to better match the pharmacokinetics of antibodies as well as long-lived therapeutic radiometals, such as, Ac-225. The manuscript provides a straightforward and systematic evaluation of the 89Zr-lintuzumab conjugate in vitro and in vivo.

Following a literature search this appears to be the first manuscript to report the evaluation of 89Zr-lintuzumab; abstracts have been presented. A report of PET imaging using another anti-CD33 antibody with Copper-64 has been reported (PMID: 31548348).

The in vitro and in vivo evaluation of the presented PET tracer is standard evaluation of a PET agent. However, the incorporation of serum stability would round out the standard evaluation. The conclusions are supported by the results and the manuscript is presented well.   The presented data would be of interest for readers working with Lintuzumab or investigating image-guided treatment for AML as well as a general interest in developing antibody-based PET tracers.  

Major

Include serum stability.

The manuscript focus on the evaluation of 89Zr-lintuzumab for the potential pre-clinical evaluation of CD33-targeted antibodies. Are there potential uses of this agent for clinical applications? How would this compare to current standards for detection of AML? Please include in introduction and discussion..

The comparison to standard PET imaging agent or shorter-lived radioisotope CD33-targeted (Ga-68 or Cu-64) agent potentially may be useful in demonstrating the advantages of the half-life of Zr-89. Pre-clinical experiments to demonstrate this may not be necessary for the current manuscript; however, adding a section with literature supporting the use of Zr-89 as compared to Ga-68, Cu-64, etc. would improve the manuscript.  

Minor Edits.

Figure 2A and B – Not sure if it’s the PDF, but please double check scale description so the writing is clear on the images.

Figure 2 C – Please add p values to graph and results if signification if counts/volume is not significant please state.

Author Response

The submitted article entitled "In vitro and in vivo characterization of 89Zirconium-labeled lintuzumab molecule” evaluates the use of Zr-89 radiometal for PET imaging of lintuzumab. The use of Zr-89 provides a longer half-life to better match the pharmacokinetics of antibodies as well as long-lived therapeutic radiometals, such as, Ac-225. The manuscript provides a straightforward and systematic evaluation of the 89Zr-lintuzumab conjugate in vitro and in vivo. – Response: We would like to thank the Reviewer for his/her encouraging opinion about our work.

Following a literature search this appears to be the first manuscript to report the evaluation of 89Zr-lintuzumab; abstracts have been presented. A report of PET imaging using another anti-CD33 antibody with Copper-64 has been reported (PMID: 31548348). – Response: We are thankful to the Reviewer for pointing out this manuscript to us, and we have included discussion of it into the revised manuscript. Please see also response to Major Comment # 3.

The in vitro and in vivo evaluation of the presented PET tracer is standard evaluation of a PET agent. However, the incorporation of serum stability would round out the standard evaluation. The conclusions are supported by the results and the manuscript is presented well.   The presented data would be of interest for readers working with Lintuzumab or investigating image-guided treatment for AML as well as a general interest in developing antibody-based PET tracers.  – Response: We completely agree with the Reviewer that stability data is important. However, the stability of DFO-Zr complex has been evaluate in many papers published before and we feel that performing such study again will not contribute new information to the field. We have added the following sentence and  three new references to the Discussion in the revised manuscript: “We did not perform the serum stability of 89Zr-DFO-lintuzumab as such stability is determined by the stability of 89Zr-DFO complex and its stability has been previously demonstrated in several studies [10-12]. Some uptake in the bone marrow observed in mice is due not to the instability of the radioconjugate but to the binding of humanized Fc part of lintuzumab to the murine FcRn receptors which is known to be much stronger for humanized than for murine antibodies [13].”

Major

  1. Include serum stability. – Response: Please see the response to the comment above.
  2. The manuscript focus on the evaluation of 89Zr-lintuzumab for the potential pre-clinical evaluation of CD33-targeted antibodies. Are there potential uses of this agent for clinical applications? How would this compare to current standards for detection of AML? Please include in introduction and discussion. – Response: We have clarified the revised Introduction that: “In the current study, we labeled lintuzumab with  89Zr and performed in vitro and in vivo characterization of this molecule with the ultimate goal to develop a preclinical tool to study CD33 tumor targeting in AML models using PET.” We have also added the following sentences to the revised Discussion:  “While AML is being successfully diagnosed with blood tests followed by bone marrow biopsies [8], an in vivo research tool to study the interaction of the anti-CD33 antibodies with the tumors would be beneficial. Here we report the results of in vitro and in vivo characterization of 89Zr-lintuzumab as a model to evaluate the in vivo binding properties of anti-CD33 antibodies in preclinical models of CD33 expressing AML using PET.”
  3. The comparison to standard PET imaging agent or shorter-lived radioisotope CD33-targeted (Ga-68 or Cu-64) agent potentially may be useful in demonstrating the advantages of the half-life of Zr-89. Pre-clinical experiments to demonstrate this may not be necessary for the current manuscript; however, adding a section with literature supporting the use of Zr-89 as compared to Ga-68, Cu-64, etc. would improve the manuscript.  – Response: We have already mentioned Ga-68 in the Introduction. We have now added the following sentence and the new reference 4 to the revised Introduction: “Recently pre-clinical imaging of acute myeloid leukemia (AML) models with Copper-64 (64Cu) has been reported [4], however, both short-physical half-life of 64Cu (12.7 hrs) and its not being readily available limit its use as a research tool for biodistribution and pharmacokinetics of antibodies.”

Minor Edits

Figure 2A and B – Not sure if it’s the PDF, but please double check scale description so the writing is clear on the images. – Response: We have improved the quality of the figures.  

Figure 2 C – Please add p values to graph and results if signification if counts/volume is not significant please state. – Response: We have added the p values and significance to the new Fig. 2C.

Reviewer 2 Report

This work describes the radiolabeling of a commercially available antibody with Zr-89 and its preliminary evaluation as potential PET imaging agent. While there is an interest in developing imaging pairs for Ac-225 labeled antibodies, the larger aim of this research project is not very clear to this reviewer. Considering that Ac-225 labelled conjugate is already in clinical trials (as mentioned in the introduction), the authors should explain how Zr-89 labelled analogue will be useful to advance these clinical studies or otherwise?

Overall, the manuscript lacks important details that can support the experimental design. CD-33 is a target that is expressed on immune cells and in different organs. This is not sufficiently discussed in the manuscript. Visually, there is an apparent discrepancy between shown PET/CT images (Figure 2) and the ex vivo biodistribution graph (Figure 3): while images show virtually no radioactivity except in tumor xenograft, especially at late time points, the biodistribution data show important/comparable accumulation in other tissues and organs (e.g. bone, lungs, spleen). This disparity is not clearly addressed in the discussion. The discussion is mainly focused on the differences in obtained radiochemical yields as compared to previously published data and does not address sufficiently biochemical aspects of the radiolabelled conjugate use (including its in vivo stability, which can be argued as problematic with high bone uptake seen from the biodistribution graph). The authors propose Zr-89 analogue to follow the faith of Ac-225 labelled antibody in vivo, but zirconium and actinium have different (coordination) chemistry and a potential impact on the conjugate properties, if any, is not clearly presented. It is a common practice to replace a blocking experiment for the evaluation of antibodies with the injection of a non-specific radiolabeled antibody. The authors should explain their choice to perform the blocking experiment and how they selected the blocking agent and dose. The observed slow clearance during blocking experiments merits additional discussion and a hypothesis. Also, since the blocking of the uptake in tumor was not fully achieved, the authors should corroborate on how significant the obtained result is. The experimental design is not very well justified and it is not clear if any statistical analysis was done (or where these results are presented / discussed).

The experimental procedures are not sufficiently detailed to be able to repeat the experiments. Some of the critical aspects that are missing include, for example, HPLC and TLC conditions for quality control, addition of DTPA during radiolabeling and its impact on the determination of the radiochemical yield (how the retention factor of Zr-89-DTPA compares to an ionic Zr-89 or the radiolabeled antibody), exact parameters for image reconstruction and image analysis.

Some jargon (“chelexed”, etc.) is present in the text of the article (which will not be understood by general readership or even radiochemists working outside of radiolabeling with metals), as well as non-conventional writing of units for specific activity (slash shall be used instead of colon). In addition, use of non-SI units such as Ci is strongly discouraged in scientific literature; SI units of radioactivity, namely Bq (MBq, GBq) shall be used instead.

The quality/resolution of PET images (at least those in generated for review pdf file) is not good enough and the quantification scale (SUV or similar) is not readily readable.

Author Response

Reviewer 2

This work describes the radiolabeling of a commercially available antibody with Zr-89 and its preliminary evaluation as potential PET imaging agent. While there is an interest in developing imaging pairs for Ac-225 labeled antibodies, the larger aim of this research project is not very clear to this reviewer. Considering that Ac-225 labelled conjugate is already in clinical trials (as mentioned in the introduction), the authors should explain how Zr-89 labelled analogue will be useful to advance these clinical studies or otherwise? – Response: We have now added the following sentence to the Discussion of the revised manuscript to clarify the goal of the study: “While AML is being successfully diagnosed with blood tests followed by bone marrow biopsies [8],  an in vivo research tool to study the interaction of the anti-CD33 antibodies with the tumors would be beneficial. Here we report the results of in vitro and in vivo characterization of 89Zr-lintuzumab as a model to evaluate the in vivo binding properties of anti-CD33 antibodies in preclinical models of CD33 expressing AML using PET.”

Overall, the manuscript lacks important details that can support the experimental design. CD-33 is a target that is expressed on immune cells and in different organs. This is not sufficiently discussed in the manuscript. – Response: As the goal of this manuscript is to characterize the 89Zr-lintuzumab molecule as a research model tool, we did not address the CD33 distribution in normal organs. This is addressed in references 5-7 which are cited in the Introduction.

Visually, there is an apparent discrepancy between shown PET/CT images (Figure 2) and the ex vivo biodistribution graph (Figure 3): while images show virtually no radioactivity except in tumor xenograft, especially at late time points, the biodistribution data show important/comparable accumulation in other tissues and organs (e.g. bone, lungs, spleen). This disparity is not clearly addressed in the discussion. – Response: The PET/CT images are reconstructed using the manufacturers algorithms to calibrate the uptake in different organs to the brightest organ which explains why the organs with the low uptake are not being visible on the images. We have added the following information on the image reconstruction and analysis to the revised Methods: “PET/CT images were registered and reconstructed automatically by the GNEXT system. PET images were reconstructed using a 3D-ordered Subset Expectation Maximization algorithm with 24 subsets and 3 iterations, CT images were reconstructed using a Modified Feldkamp algorithm. Reconstructed PET images were filtered using gaussian smooth 3D FWHM 2x2x2mm and analyzed using p-MOD v3.903 (Zurich, Switzerland). “

The discussion is mainly focused on the differences in obtained radiochemical yields as compared to previously published data and does not address sufficiently biochemical aspects of the radiolabelled conjugate use (including its in vivo stability, which can be argued as problematic with high bone uptake seen from the biodistribution graph). – Response: We have included the following information and new references on the stability question into the Discussion: “We did not perform the serum stability of 89Zr-DFO-lintuzumab as such stability is determined by the stability of 89Zr-DFO complex and it has been previously demonstrated in several studies [10-12]. Some uptake in the bone marrow observed in mice is due not to the instability of the radioconjugate but to the binding of humanized Fc part of lintuzumab to the murine FcRn receptors which is known to be much stronger for humanized than for murine antibodies [13].”

The authors propose Zr-89 analogue to follow the faith of Ac-225 labelled antibody in vivo, but zirconium and actinium have different (coordination) chemistry and a potential impact on the conjugate properties, if any, is not clearly presented. – Response: As we stated in the Introduction and at the beginning of the Discussion, we are not proposing to use 89Zr-lintuzumab for the pre-selection of patients before treating them with 225Ac-lintuzumab but to use it as a tool in pre-clinical work to evaluate with PET interaction of CD33-binding antibodies with tumor cells.

 It is a common practice to replace a blocking experiment for the evaluation of antibodies with the injection of a non-specific radiolabeled antibody. The authors should explain their choice to perform the blocking experiment and how they selected the blocking agent and dose. The observed slow clearance during blocking experiments merits additional discussion and a hypothesis. Also, since the blocking of the uptake in tumor was not fully achieved, the authors should corroborate on how significant the obtained result is. The experimental design is not very well justified and it is not clear if any statistical analysis was done (or where these results are presented / discussed).  – Response: We agree with the Reviewer that using a non-specific antibody is one way to demonstrate the specificity of the antibody of interest. However, there is a second, also well tested way to show the specificity through blocking of the specific binding sites by using the excess of unlabeled reagent. The statistical significance of the results is described in the Results on page 4: “SUV analysis of tumor images taken on Days 3 and 7 showed a significantly (p=0.001 and p<0.0001, respectively) higher uptake of 89Zr-DFO-Lintuzumab in the tumors of non-blocked mice than in the pre-blocked tumors (Fig.2C). In addition, Time-Activity Curves (TAC) show that the radiolabeled antibody remains mostly constant over the 7 day period after initial uptake for both non-blocked and pre-blocked tumors, with the pre-blocked has significantly less uptake (Fig. 2D). “ In addition, we have added the reference to this methodology in the revised Discussion: “Pre-blocking of uptake with the unlabeled specific molecule administration is a well tested way to demonstrate the specificity of a molecule for its receptor or an antigen [14]. Analysis of the time activity curve (TAC) shows that there is only a slight downwards slope indicating that the majority of the 89Zr that accumulates in the tumor remains present there until decay.”

The experimental procedures are not sufficiently detailed to be able to repeat the experiments. Some of the critical aspects that are missing include, for example, HPLC and TLC conditions for quality control, addition of DTPA during radiolabeling and its impact on the determination of the radiochemical yield (how the retention factor of Zr-89-DTPA compares to an ionic Zr-89 or the radiolabeled antibody), exact parameters for image reconstruction and image analysis. – Response: We have added the following information to the revised Methods: “iTLC were eluted using a 50mM EDTA mobile phase with the radiolabeled antibody remaining at the point of application (Rf=0) and the DTPA bound 89Zr moves with the solvent front (Rf=1), the strip was then cut in half and the measured individually, allowing for calculation of radiolabeling yields. SEC-HPLC traces were performed using a Tosoh BioScience (Tokyo, Japan) TSKgel UP-SW-3000 SEC column with an isocratic method using phosphate buffered saline (PBS) as a mobile phase. UV and Radio traces were collected and the peak area compared to iTLC results, only a single radio peak that corresponded with the antibody UV peak was observed. Four mice were scanned for 10 minutes (static) simultaneously using a multi-mouse bed.  PET/CT images were registered and reconstructed automatically by the GNEXT system. PET images were reconstructed using a 3D-ordered Subset Expectation Maximization algorithm with 24 subsets and 3 iterations, CT images were reconstructed using a Modified Feldkamp algorithm. Reconstructed PET images were filtered using gaussian smooth 3D FWHM 2x2x2mm and analyzed using p-MOD v3.903 (Zurich, Switzerland).  Volume of interest (VOI) analysis was performed using p-MOD. MIP images were generated to help visualize tumor uptake. Tumor VOI were drawn manually, activity was decay corrected. Standardized uptake values (SUV) were calculated using the equation SUV = C/(dose/weight) where C is the tissue radioactivity concentration, weight is weight of mouse and dose is injected dose of radioactive antibody.”

Some jargon (“chelexed”, etc.) is present in the text of the article (which will not be understood by general readership or even radiochemists working outside of radiolabeling with metals), as well as non-conventional writing of units for specific activity (slash shall be used instead of colon). – Response: We have replaced these words with the following: “The desired amount of 89Zr(Ox)2 in 1M oxalic acid was dissolved in 0.5M HEPES buffer that had been run through a chelex cation exchange resin to remove any advantageous metals and neutralized using 1M Na2CO3.” The colons have been replaced.

In addition, use of non-SI units such as Ci is strongly discouraged in scientific literature; SI units of radioactivity, namely Bq (MBq, GBq) shall be used instead. – Response: We have replaced the non-SI with SI units.

The quality/resolution of PET images (at least those in generated for review pdf file) is not good enough and the quantification scale (SUV or similar) is not readily readable. – Response: Images were updated to higher resolution.

Reviewer 3 Report

interesting study with 89Zr-conjugated with lintuzumab. definitely as the authors mentioned, the longer half life of the zirconium provides greater flexibility for the study. introduction can be more specific and add more information related to study. fig 2. images look good, how the tumors were confirmed before PET, was there a CT or MRI done. fig. 2. In addition to the above comment, the images were coregistered with the CT or MRI images. authors can add a TAC with baseline and blocking studies. fig 3. Please mention n = ? for biodistribution and imaging. fig 3. add the SUV value anywhere possible and compare as the data should be already available. page 6, line 196 "ug"

Author Response

Interesting study with 89Zr-conjugated with lintuzumab. definitely as the authors mentioned, the longer half life of the zirconium provides greater flexibility for the study. – Response: We would like to thank the Reviewer for his/her encouraging comments about the work.

Introduction can be more specific and add more information related to study. – Response: We have added the following information to the Introduction: “. Recently pre-clinical imaging of acute myeloid leukemia (AML) models with Copper-64 (64Cu) has been reported [4], however, both short-physical half-life of 64Cu (12.7  hrs) and its not being readily available limit its use as a research tool for biodistribution and pharmacokinetics of antibodies.  It has been demonstrated that targeting CD33 with lintuzumab antibody radiolabeled with 225Ac has promising activity in patients with relapsed/refractory  AML in phase 1/2 clinical trials [5-7]. In the current study, we labeled lintuzumab with  89Zr and performed in vitro and in vivo characterization of this molecule with the ultimate goal to develop a preclinical tool to study CD33 tumor targeting in AML models using PET.”

 fig 2. images look good, how the tumors were confirmed before PET, was there a CT or MRI done. – Response: The tumors were xenografted into the flank, so they were visually confirmed.

fig. 2. In addition to the above comment, the images were coregistered with the CT or MRI images. – Response: PET was co-registered with CT, we added this to the caption of Fig. 2.

authors can add a TAC with baseline and blocking studies. fig 3. Please mention n = ? for biodistribution and imaging. fig 3. add the SUV value anywhere possible and compare as the data should be already available. – Response: SUV and TAC data have been generated and added to Fig. 2 as new Fig. 2C and 2D. The number of mice (4 per group) is present in the captions of Fig. 2 and Fig. 3.

page 6, line 196 "ug" – Response: Have been corrected.

Round 2

Reviewer 2 Report

The text of the manuscript has been improved by adding a number of essential details, including to the experimental section. Still, there are a few outstanding points that should be clarified or expanded upon:

- A sentence has been added to the Discussion (“While AML is being successfully diagnosed with blood tests followed by bone marrow biopsies [8], an in vivo research tool to study the interaction of the anti-CD33 antibodies with the tumors would be beneficial."), hinting on potential benefits of a developed in vivo research tool. The authors should expand on why this "would be beneficial" providing some concrete examples.

- An apparent difference between the results of PET/CT on Figure 2 and the ex vivo biodistribution in Figure 3 are still not clear to this reviewer. Looking at the blocking experiment results (Figure 3), the radioactivity in the tumor, spleen, bone, and lungs are approximately at the same level (visually some 7-10%) and blood radioactivity is even higher. On the PET/CT images however, the tumor is clearly and almost exclusively visible. What was the tumor-to-blood and tumor-to-muscle ratios that was found for PET and how they compare to those obtained by ex vivo distribution? If the results obtained with PET and with the biodistribution differ significantly, the authors should provide a reasonable explanation.

- Were the images in Figure 2 normalized to the same scale in terms of signal (kBq/cc or SUV) across different days to show an evolution of tracer uptake in tumor? In addition, the authors should verify if the scale bars on PET images are indeed show correct units (kBq/cc) since SUV calculation method is mentioned elsewhere in the manuscript. 

- The authors observed slow clearance of the radiolabeled antibody during blocking experiments but the possible reasons for this are not sufficiently addressed in the discussion.

- For statistics, the authors should mention if they performed a power analysis to calculate their sample size for animal studies (and how) or otherwise include a statement justifying the sample size.

Author Response

The text of the manuscript has been improved by adding a number of essential details, including to the experimental section. Still, there are a few outstanding points that should be clarified or expanded upon:

- A sentence has been added to the Discussion (“While AML is being successfully diagnosed with blood tests followed by bone marrow biopsies [8], an in vivo research tool to study the interaction of the anti-CD33 antibodies with the tumors would be beneficial."), hinting on potential benefits of a developed in vivo research tool. The authors should expand on why this "would be beneficial" providing some concrete examples. – Response: We have added the following example to the revised Discussion: “For example, the timing of an anti-CD33 antibody uptake, retention and washout from the tumor will determine the timing of administration of an additional anti-cancer agent(s) for combination therapy to achieve a synergistic effect.”

- An apparent difference between the results of PET/CT on Figure 2 and the ex vivo biodistribution in Figure 3 are still not clear to this reviewer. Looking at the blocking experiment results (Figure 3), the radioactivity in the tumor, spleen, bone, and lungs are approximately at the same level (visually some 7-10%) and blood radioactivity is even higher. On the PET/CT images however, the tumor is clearly and almost exclusively visible. What was the tumor-to-blood and tumor-to-muscle ratios that was found for PET and how they compare to those obtained by ex vivo distribution? If the results obtained with PET and with the biodistribution differ significantly, the authors should provide a reasonable explanation. – Response: The PET/CT images were originally processed using MIP starting with their initial values at the time of image collection and were used to visualize tumor uptake with blood pool being thresholded out. Upon the Reviewer’s excellent suggestion we have reprocessed the images showing SUV and normalized it to the same scale, we also reduced the cut off threshold to show blood pool 89Zr-lintuzumab, so it allows the reader to see that the biodistribution and imaging results closely align with each other.  The reprocessed images are shown in revised Fig. 2A and 2B. The tumor to blood and tumor to muscle ratios were in agreement between  PET and biodistribution on day 7 – for example, in pre-blocked mice tumor to blood ratio by  PET was 0.72 and by biodistribution - 0.65, tumor to muscle ratio by PET was 4.7  and by biodistribution – 4.0.  We have included this sentence into the revised Discussion.

- Were the images in Figure 2 normalized to the same scale in terms of signal (kBq/cc or SUV) across different days to show an evolution of tracer uptake in tumor? In addition, the authors should verify if the scale bars on PET images are indeed show correct units (kBq/cc) since  SUV calculation method is mentioned elsewhere in the manuscript. – Response: The images were reprocessed and normalized to SUV using the same scale. We have added this information to the revised Methods.

- The authors observed slow clearance of the radiolabeled antibody during blocking experiments but the possible reasons for this are not sufficiently addressed in the discussion. – Response: We have added the following explanation to the revised Discussion: “The slower clearance of the radiolabeled antibody from the circulation in pre-blocked animals can be explained by its inability to bind in high amount to the pre-blocked tumor (which usually serves as an antibody “sink”), thus leaving the majority of the radiolabeled antibody molecules in the circulation for a long time.”

- For statistics, the authors should mention if they performed a power analysis to calculate their sample size for animal studies (and how) or otherwise include a statement justifying the sample size. – Response: We have created a Statistical Analyses section in the revised manuscript which now contains the following information: “Power analysis for the  PET/CT and biodistribution studies was estimated using PASS version 11 (NCSS, Inc.) using simulations of different tumor uptakes based on pilot data and conservative assumptions regarding the groups pre-blocked with unlabeled lintuzumab. All simulations showed power of at least 80% with only four animals per group because of the large differences between uptake in pre-blocked and non-blocked mice. Thus, 4 mice per group were utilized in the in vivo studies. “